

# Coral reef degradation at an atoll of the Western Colombian Caribbean

Natalia Rivas[1], Carlos E. Gómez[1,2], Santiago Millán[1], Katherine Mejía-Quintero[1] and Luis Chasqui[1]

[1] Programa de Biodiversidad y Ecosistemas Marinos, Instituto de Investigaciones Marinas y Costeras-INVEMAR, Santa Marta, Colombia
[2] Laboratorio de Biología Molecular Marina-BIOMMAR, Universidad de los Andes, Bogotá, Colombia

## ABSTRACT

Coral reef decline is an issue of concern around the globe. Remote and uninhabited coral areas are not exempt from facing changes in species composition and functionality due to global drivers. Quitasueño is a remote atoll within the Seaflower Biosphere Reserve, in the Southwestern Caribbean Sea. To evaluate the current status of the coral reefs in Quitasueño we sampled 120 stations through Rapid Ecological Assessment and evaluated four stations through Planar Point Intercept to compare the current percent cover of benthic groups with previous studies in the area. We found pronounced changes in coral and macroalgae covers in time, and great conspicuousness of multiple conditions of deterioration along Quitasueño, including diseases, coral predation, and aggression and invasion of coral colonies by macroalgae and sponges. The reef ecosystem seems to be facing a phase shift, in which the benthic cover previously dominated by hard corals is currently dominated by fleshy macroalgae. It is essential to evaluate the possible drivers of the extent of degradation of Quitasueño to understand the process of deterioration and mitigate the impacts.

## INTRODUCTION

Coral reefs are facing multiple threats, which have caused great deterioration and loss of ecosystem functionality worldwide for the last quarter-century (*Burke et al., 2011*; *Souter et al., 2021*). Some threats work on a global scale related to climate change, such as coral bleaching due to increase in sea surface temperature (*Brown et al., 2019*); others operate at the local or regional levels related to local human activities, such as pollution, terrestrial runoff, and overfishing (*McLean et al., 2016*). In that sense, lower levels of coral decline are usually expected in remote and uninhabited areas due to the lesser direct influence of human activities, however, coral degradation in such areas has also been reported (*Coelho & Manfrino, 2007*; *Gardner et al., 2003*), with remarkable changes in the benthic community composition as a common indicator.

Heavily degraded reefs typically exhibit low cover of reef-building corals, with fleshy macroalgae dominating the bottom (*Perry et al., 2018*). In addition, shifts in the dominance of those reef-building corals are also associated with degradation processes.

Corresponding author
Natalia Rivas,
natalia.rivas@invemar.org.co

In the western Atlantic, the branching acroporids used to be the main reef builders and habitat providers on shallow waters, however, due to their massive mortality in the 80s and 90s, *Orbicella* species (massive corals) became the main reef-forming corals (*Estrada-Saldívar et al., 2019*). The remaining populations of *Acropora* spp. and other reef-building species have decreased across the region due to emergent diseases and other environmental stressors, which have favored fast growing non-framework coral species such as *Porites astreoides* and *Agaricia agaricites* (*Green, Edmunds & Carpenter, 2008*; *Estrada-Saldívar et al., 2019*). Therefore, the assessment and monitoring of coral reef conditions and the drivers of its decline in continental and oceanic localities are crucial to understand ecosystem dynamics, addressing possible ways to mitigate stressors, and designing recovery strategies when necessary (*Sánchez et al., 2019*).

In the Colombian Caribbean, the Archipelago of San Andrés, Providencia, and Santa Catalina, nominated as the Seaflower Biosphere Reserve since 2000, comprises several remote and oceanic islands, atolls, cays, and shoals with almost 80% of the coral reefs areas of the country (*Abril-Howard et al., 2012*). The reserve includes the Marine Protected Area Seaflower (MPA Seaflower) whose northern section (37,522 km$^2$) holds multiple submerged banks and the three northern atolls of Roncador (13°34 N, 80°5 W), Serrana (14°17 N, 80°21 W), and Quitasueño (14°20 N, 81°11 W) (*Taylor et al., 2012*). The latter is a volcanic basement atoll (*Geister, 1975*) with no inhabited land, and harbors the largest coral area in the country with over 1,300 km$^2$ (*Millán & García-Valencia, 2021*). Within the types of uses established inside the MPA Seaflower, Quitasueño is divided into non-take and artisanal fishing zones (*Taylor et al., 2012*). Few studies have evaluated the benthic community of the area for its remoteness.

Understanding vulnerability as the probability of detrimental effects due to the exposure to degradation conditions, *Barrios (2000)* described a high vulnerability of Quitasueño not only due to the high frequency of conditions but also to the great frequency of healthy coral colonies, susceptible to be affected by those conditions. In general, the study reported that Quitasueño had high frequency of current mortality, high levels of coral predation by mobile organisms (fish, gastropods, and fire worms), high ratios of invasions (covering and smothering), and aggression (contact) by macroalgae, sponges and other organisms like octocorals, zoanthids, and tunicates including ascidians. In addition, Quitasueño displayed a high frequency of physical damage on coral reefs such as overturns, scratches or scrapings, and sedimentation. Further, the seven assessed diseases (yellow band, white band, black band, red band, white spots, dark spots, and white patches) were reported in high frequency at the locality.

The great healthy coral cover was later confirmed by *Sánchez et al. (2005)*, with data collected using the Planar-Point Intercept (PPI). The evaluation of the benthic communities in the three northern atolls of Serrana, Roncador, and Quitasueño, indicate that Quitasueño had the greatest mean coral cover (33%) ranging between 18% to 50%, and of fleshy macroalgae between 4% to 39%. They also described spatially structured habitats in the atoll with differences between North and South of the complex. The most recent information about the benthic community of the area (*Abril & Arango, 2012*) confirms the ongoing decline indicating lower coral cover than reported by *Sánchez et al.*

*(2005)*, however, methods and benthic categories included in the analysis were not comparable. This was also suggested in a study of macro-algae diversity of Quitasueño, where a rather low coral cover was reported as well, but no quantitative data were offered to support that assertion, instead extensive cyanobacterial mats were described as a possible indicator of coral decline (*Gavio, Cifuentes-Ossa & Wynne, 2015*).

More than 20 years after the first quantitative assessments of coral reef degradation and vulnerability in Quitasueño (*Barrios, 2000*) we surveyed the area as part of a scientific expedition to five oceanic reef complexes of the western Colombian Caribbean to update the maps of the shallow marine ecosystems. In Quitasueño the high level of deterioration of coral reefs was conspicuous in relation to the other reef complexes. Here we present the results of a video-transects based analysis and Rapid Ecological Assessment, and a brief description of several signs of coral decline, as an update of the general health status of the coral ecosystems in the atoll, but mainly as a warning on the urgent need for an in-deep ecosystem health assessment in this and other remote coral areas of the Southern Caribbean.

## MATERIALS AND METHODS

### Study area and data collection

Quitasueño is an elongated atoll of 63 km long and 23 km wide (*Millán & García-Valencia, 2021*). The reef system has an extensive leeward fore reef dominated by scleractinian corals towards the mid part, and large octocoral gardens and sponges dominating reef bottoms towards the north and south segments of the complex (*Millán & García-Valencia, 2021*). Historically, the lagoon basin was dominated by ecological units of *Orbicella spp*. and *Acropora palmata-Pseudodiploria strigosa*, with the former also dominating the fore reef (*INVEMAR-MINAMBIENTE, 2020*).

During November 2021, we evaluated 120 sampling stations through Rapid Ecological Assessment (REA) on coral-reef bottoms of Quitasueño Bank (14°20 N, 81°11 W) during the Seaflower Plus scientific expedition, which was part of a governmental effort to update the habitat maps and to identify priority areas for restoration and conservation in the Seaflower Biosphere Reserve. To describe the current benthic community of Quitasueño we used two complementary approaches. A spatial approach involved the visual assessment of the percent cover of benthic groups, and a temporal approach aimed to identify changes in the current benthic cover compared to previous studies. For the later approach, we selected *Sánchez et al. (2005)* to produce comparable data in time, due to a more detailed description made by these authors about the fieldwork and data analysis procedures they followed, allowing us to replicate their methods to a certain extent.

### Spatial scale

We used REA to record information on the percent cover of shallow benthic communities (<30 m) following similar methods used by *Díaz, Garzón-Ferreira & Zea (1995)*, *Garzón-Ferreira & Pinzón (1999)* and *Andréfoüet & Guzman (2005)*. The benthic groups included hard corals (C; Scleractinia and Milleporidae), fleshy macroalgae (MALG), calcareous macroalgae (CALG), encrusting algae (EALG), Cyanophyta algae (CYAN), octocorals
(OCTO), sponges (SPG), and zoanthids (ZOAN). During REA, two researchers visually estimated the percent cover of the benthic community and photographed the most conspicuous benthic organisms in a radius of 6 m around a marking point. The estimated percentages made by each researcher were averaged and registered per station. Surveys included sites in lagoon basin, back reef, and fore reef attempting to cover the largest possible area and geomorphological traits of the reef complex. Each station was georeferenced with a GPS and stations were classified according to depth as follows: shallow (<8 m), medium (8–12 m), and deep (>12 m).

## Statistical analysis

The estimated percentages of benthic organisms considered during the REAs were standardized to 100% and transformed to arcsine root (*Gotelli & Ellison, 2004*).

To ascertain how stations grouped depending on the visual percentages estimated in the field and identify structure in our data, we performed a CLUSTER analysis based on a Bray-Curtis similarity matrix and SIMPROF test following *Somerfield & Clarke (2013)*. The groups of stations formed in the CLUSTER analysis were considered as a new factor of the data, which was then used to perform a non-metric multidimensional scaling ordination analysis (nMDS) and a SIMPER analysis to establish which benthic groups contributed the most to the clustered stations (*Clarke, 1993*). Finally, to visually assess how the primary benthic groups were spatially arranged, we generated three maps; one with georeferenced stations classified according to the cluster results, and two bubble maps to spatially represent the percent cover of the fleshy macroalgae and hard corals across the bank. All the statistical analysis and figures were performed in Primer (v.6.1.1.4), ArcGIS (v. 10.8), and R i386 (v. 4.1.3; *R Core Team, 2022*).

## Temporal scale

We surveyed four stations with video transects using the Planar-Point-Intercept method (PPI) (*Dodge, Logan & Antonius, 1982*). In the field, 10 m transects were placed at each station and the same diver recorded the benthic composition with a video camera. Transects were conducted in depths between 3 to 19 m and were located on the back reef, lagoon basin, and fore reef. The video processing consisted of creating quadrants at each meter of both sides of the transects using the metric tape as a reference and a grid of points separated every 10 cm, trying to closely replicate the methods described by *Sánchez et al. (2005)*. As a result, 6,070 points were evaluated, which was equivalent to 607 linear meters with benthic cover evaluated every 10 cm, in contrast to the 7,413 linear meters evaluated by *Sánchez et al. (2005)* in the 48 stations considered. In addition, we used the same benthic categories used by *Sánchez et al. (2005)* including hard corals (Scleractinia and Milleporidae), fleshy macroalgae (Macroalgae), calcareous macroalgae, encrusting calcareous algae, filamentous algae, octocorals, erect sponges, and encrusting sponges. Unlike *Sánchez et al. (2005)*, the category "Other fleshy invertebrates" was not included in this study.

   To compare the percent cover among benthic communities back in 2003 and the observed in 2021, we generated a similar boxplot chart to the one produced by *Sánchez*

*et al. (2005)*. Since we only evaluated four stations with PPI, we made a similar graph with percentage data obtained in the 120 stations evaluated through REAs to complement the results obtained through PPI. Although it was a different method than PPI (*e.g.*, *Sánchez et al., 2005*), we decided to include the box plot with the data obtained through REA because we cover the same habitat types (lagoon basin, back reef, and fore reef) and depths (0–30 m) (refer to Fig. 5A in *Sánchez et al. (2005)* to see the stations surveyed). However, caution is required in the interpretation of this data as methods and sampling efforts differs between both studies.

## Conditions of degradation

We evaluated photographs taken at each station to semi-quantify the conditions of degradation within the complex. Such conditions included: (1) Current mortality; considered whenever the coral tissue had recently died but the coral skeleton was visible and identifiable to species level, (2) old mortality; when coral structures were covered with non-easily removed organisms like sponges and algae; predation by mobile organisms (gastropods), (3) invasions; when corals were covered or smothered by sponges or algae, (4) aggression; when corals were in contact with algae, (5) presence of diseases, and (6) physical damage (turnover, scratches, and sedimentation). We warn about the possible underestimation of the condition of degradation as they are dependent on whether they were observed and photographed by the field researchers at each station.

# RESULTS

## Spatial scale

Stations were clustered into 11 groups based on the benthic coverage estimated. The most frequently observed coral species were *Siderastrea siderea* (82%), followed by *Porites astreoides* (71%), *Agaricia agaricites* (67%), *O. annularis* (56%), and *O. faveolata* (53%).

Algae and hard corals were the dominant benthic organisms except for group I, with octocorals as the dominant component (Fig. 1). This was supported by the nMDS, which showed hard corals and fleshy macroalgae as the benthic groups largely involved in the ordination of stations (Fig. S1). Furthermore, in the group with the highest number of stations (Group B = 50) the SIMPER analysis showed that fleshy macroalgae contributed to over 78% of the similarity between stations (Figs. 1–3A, Table S1). Group A was second in the number of stations (22), with fleshy macroalgae contributing mostly to the similarity among stations (58%), and a greater contribution of coral cover (24.08%) setting them apart from group B. Nine more stations showed the highest coral cover, contributing to over 84% of the similarity among stations (group J, Fig. 1).

Once the stations were represented on the map according to the groups formed by the CLUSTER analysis, the configurations of the benthic community composition did not show to be spatially arranged along the reef complex (Fig. 2A). For example, stations from group B extended through the reef complex, covering habitats from the lagoon basin, back reef, and fore reef. Likewise, the nine stations with a high coral cover (group J) stretched along the complex near the reef crest, inside the lagoon basin, and on the fore reef. Moreover, we did not find a spatial pattern once we independently plotted the percent

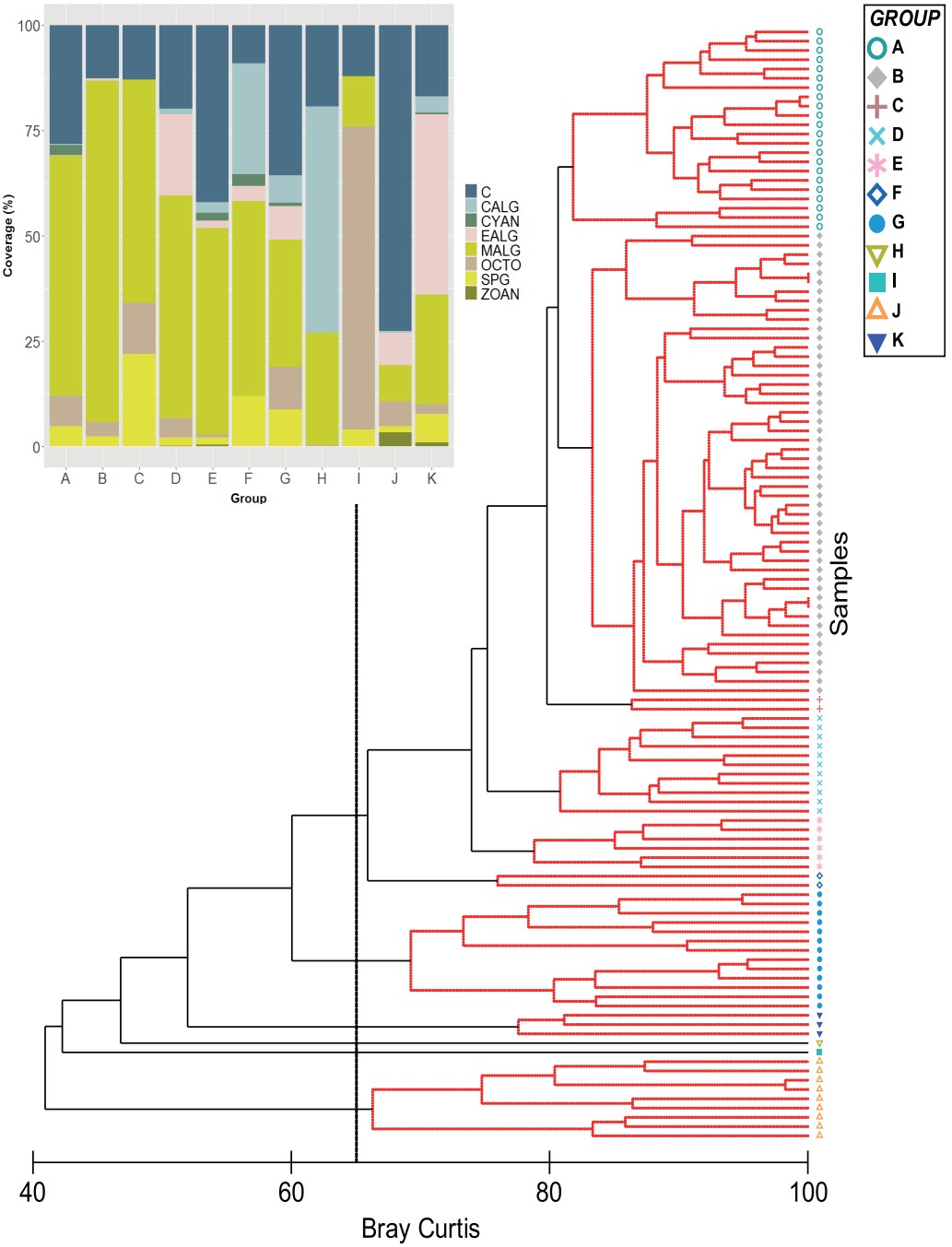

**Figure 1 Classification analysis of the benthic community of Quitasueño.** CLUSTER of similarity (Bray-Curtis) of 120 sampling stations in Quitasueño Bank (the 65% similarity line between groups is shown). The stacked bar chart shows the average cover for every benthic class. Hard corals (C; Scleractinia and Milleporidae), fleshy macroalgae (MALG), calcareous macroalgae (CALG), encrusting algae (EALG), cyanophyta algae (CYAN), octocorals (OCTO), sponges (SPG) and zoanthids (ZOAN).

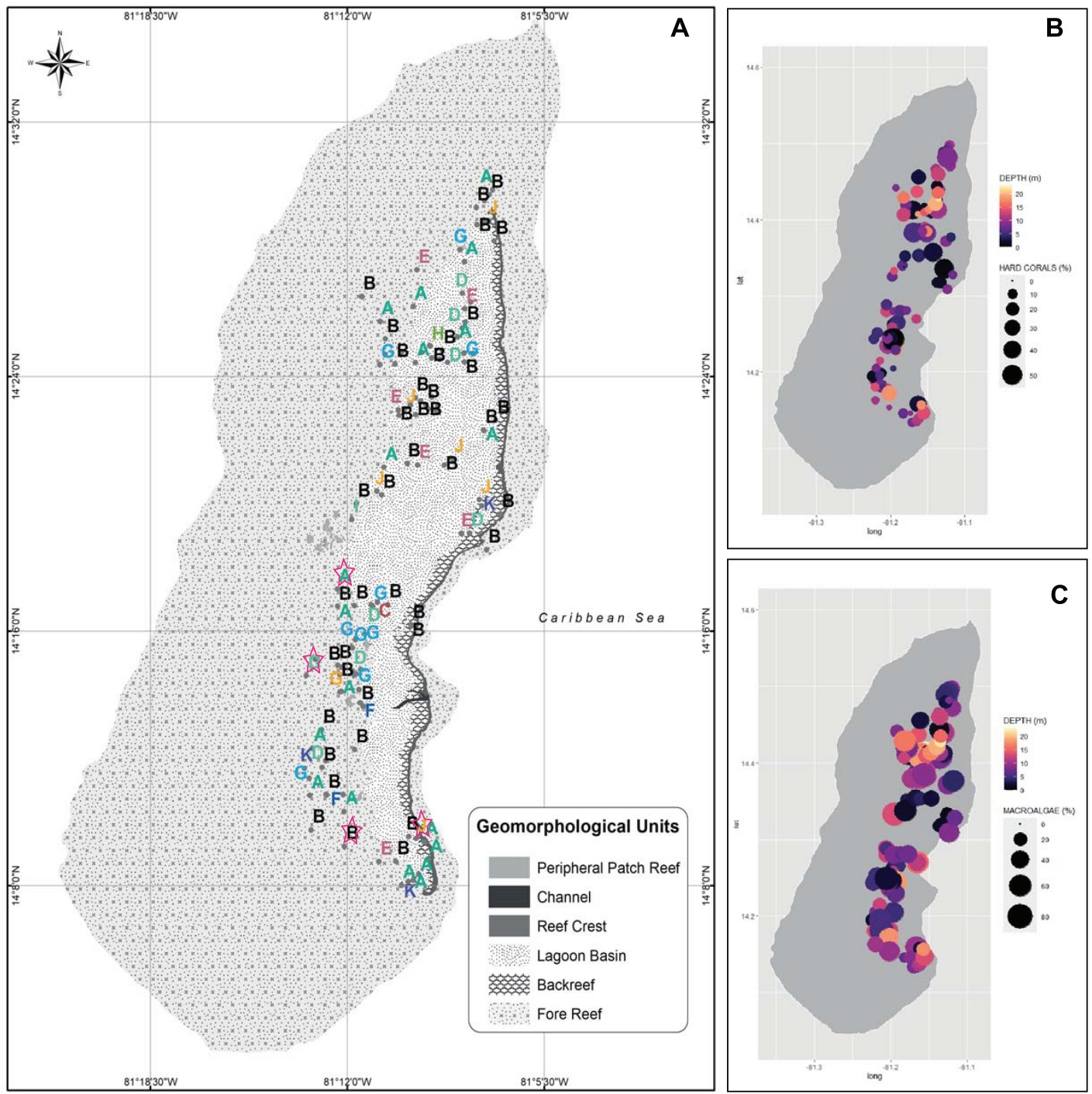

**Figure 2 Spatialized data of the benthic community of Quitasueño.** (A) Georeferenced sampling stations evaluated by rapid ecological assessment (REAs) in Quitasueño Bank. The stations are labeled according to the CLUSTER classification in Fig. 1. The legend details the geomorphological units, and the four stations evaluated through planar-point-intercept are highlighted with a star. (B and C) Bubble maps with the spatial location of the stations evaluated through REAs. Bubbles sizes represent the percentage of contribution of hard corals (B), and macroalgae (C) and panel color shows the depths of the stations.

coverages of the hard corals and fleshy macroalgae (Figs. 2B and 2C). This indicates that there was not a specific area of the complex with higher percentages of either one of the dominant benthic groups. In one hand, a high percent cover of fleshy macroalgae was found throughout the sampled area independently of the habitat type (*i.e.*, exposed habitats in the fore reefs and protected habitats in the lagoon basin), and the different depths evaluated. On the other hand, the few places where we found a high percentage of hard corals, were spread throughout the area also distributed in exposed and protected habitats, however, it only covered depths shallower than 12 m.

## Temporal scale

The comparison among benthic covers in Quitasueño surveyed by *Sánchez et al. (2005)* and those surveyed in the present study, suggest an ongoing shift in community composition within the atoll in the last two decades. Previously, corals appeared as the dominant benthic group with the greatest percent cover (median around 30%) above fleshy macroalgae (~20%) (*Sánchez et al., 2005*; Fig. 3A). Our results showed that fleshy macroalgae increased up to ~65% and was the benthic group with the greatest percent cover in 2021, high above the hard corals with a percent coverage median of around 10% (Fig. 3B). We found similar results when analyzing the percent cover of the benthic groups through REAs (Fig. 3C). Although, for methodological reasons the categories differ from those considered in the four stations evaluated through PPI, some general considerations can be made, such as hard corals and fleshy macroalgae as the most common groups in the benthic community. Likewise, the distribution of percentages in both groups was similar, with most macroalgae cover ranging between 40% to 80%, including some stations with up to 100% coverage and some others with almost no fleshy macroalgae. The median percentage of hard coral cover (~20%) did seem to differ with the stations evaluated through the PPI (~10%). However, it is still 10% lower than what was reported by *Sánchez et al. (2005)* (Fig. 3A).

## Conditions of degradation

The semi-quantification of conditions of degradation showed that Old Mortality and Aggression by macroalgae were the most frequently observed conditions with 95% of stations presenting them. Physical Damage was present in 87.5% of stations with turnovers, scratches, and holes that looked as if scoops of ice cream had been scooped out (Fig. 4). These conditions were followed by Invasions (65.8%), Current Mortality (51.7%), and Diseases (33%), including white plague, and dark spot diseases, among others (Fig. 5). Finally, Predation by Mobile Organisms was observed in 10 stations (8.3%) as only predation by gastropods was included, usually *Cyphoma gibbosum* in octocorals (Table 1).

## DISCUSSION

Quitasueño reef complex goes through a decay process, with a sharp decline in coral cover compared to previous studies (*e.g.*, *Barrios, 2000*; *Sánchez et al., 2005*). More than 20 years after the study by *Barrios (2000)*, the steady decline in Quitasueño coral reefs seems evident. According to our observations on 120 stations evaluated along the reef complex,

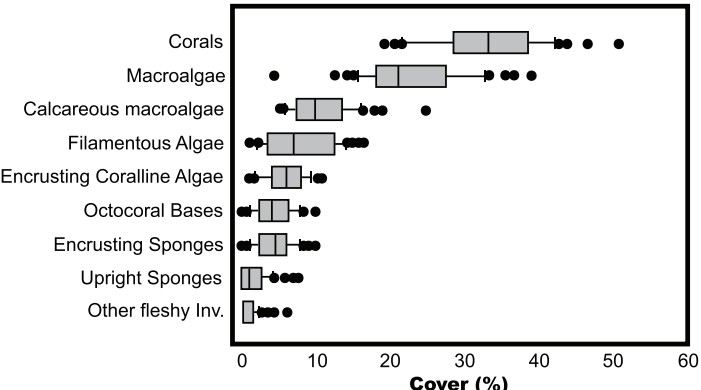

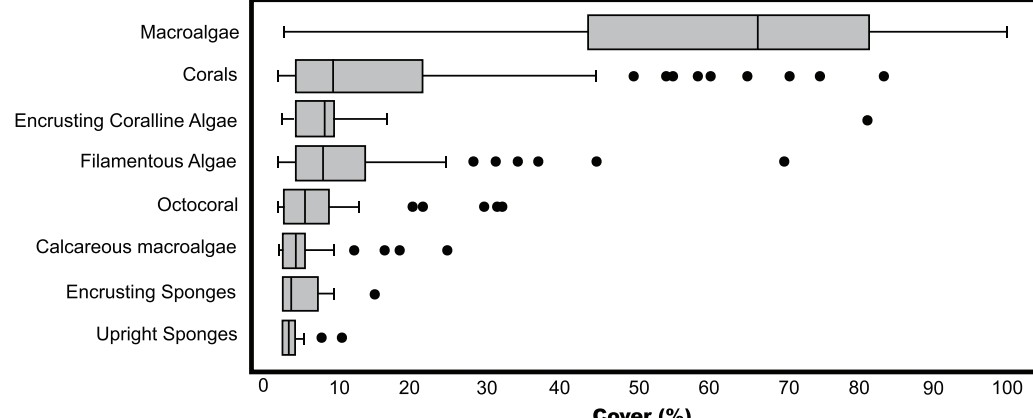

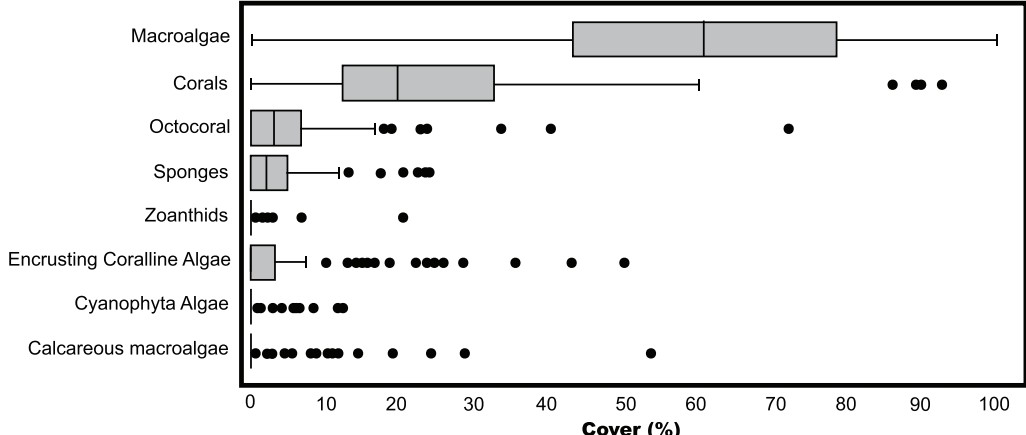

**Figure 3 Cover percentage of the benthic groups of Quitasueño in 2000 and 2021.** (A) Boxplot adapted from *Sánchez et al. (2005)* (CC BY-NC 4.0). (B) Data from video-transects and planar point intercept (PPI). (C) Data from rapid ecological assessment (REA), both collected during the seaflower plus scientific expedition 2021.

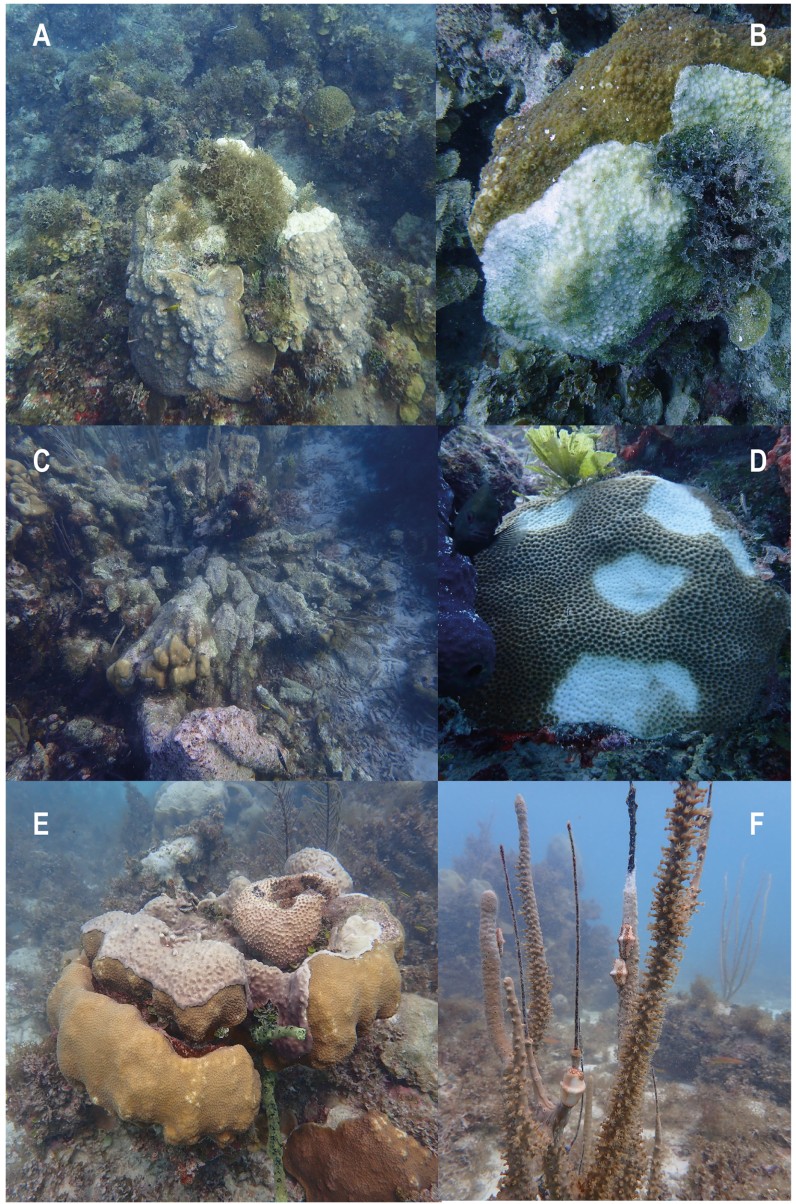

**Figure 4  Evidence of conditions of degradation at coral reefs of Quitasueño.** (A and B) Old mortality, current mortality, and aggression by macroalgae. (C and D) Physical damage. (E) Invasions. (F) Predation. Photos were taken during the seaflower plus scientific expedition in November 2021.

we can claim it is currently dominated by macroalgae since only nine stations (7.5%) had a higher coral cover compared to the other benthic categories evaluated. Moreover, the lack of spatial distribution patterns according to the benthic community composition and the percentages of hard corals and fleshy macroalgae, leads us to the belief that the degradation process is widespread throughout the complex and not something that is happening in a specific area or habitat.

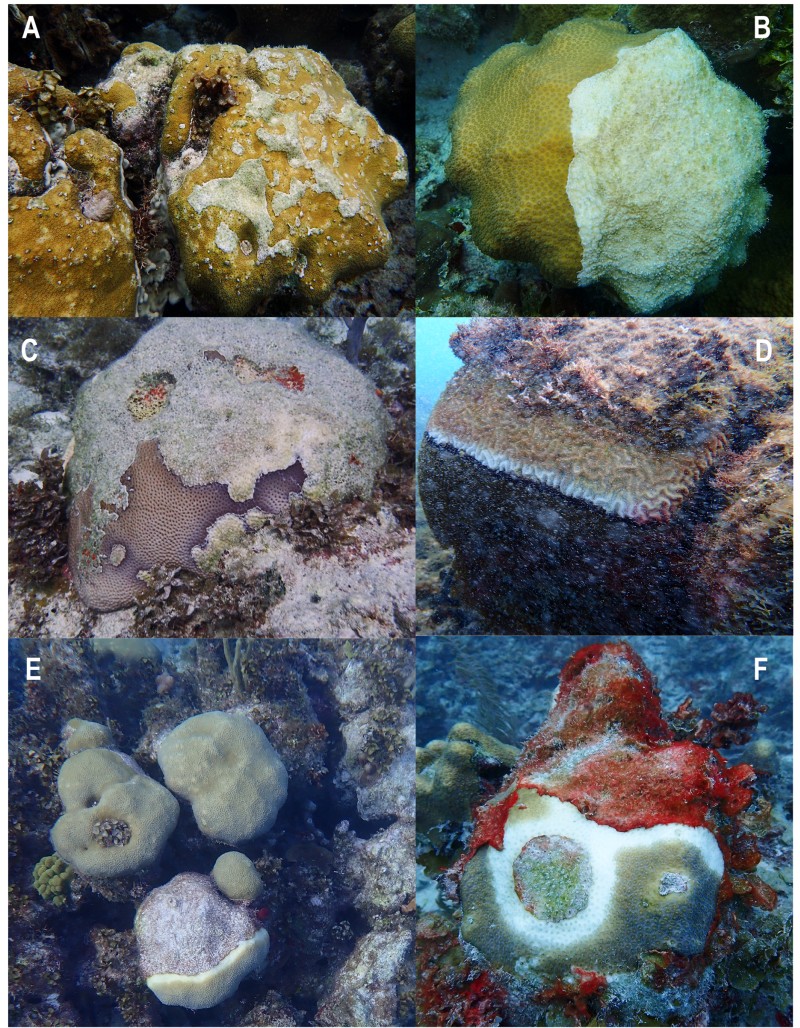

**Figure 5 Evidence of current mortality and diseases.** (A) The current mortality is possibly due to sediment damage. (B) The current mortality is possibly due to white plage. (C) Black band disease. (D) Dark spot disease. (E and F) Caribbean yellow blotch disease.

**Table 1 Semi-quantification of conditions of degradations in Quitasueño.**

| Conditions of degradation | No. of stations | Percentage |
|---|---|---|
| Current mortality | 62 | 51.7 |
| Old mortality | 115 | 95.3 |
| Predation by mobile organisms | 10 | 8.3 |
| Invasions | 79 | 65.8 |
| Aggression | 114 | 95 |
| Diseases | 33 | 27.5 |
| Physical damage | 105 | 87.5 |

**Note:**
The number and percentage of stations presenting each condition are included.

Few stations with no apparent spatial pattern of distribution remained healthy and surrounded by areas with high percentages of fleshy macroalgae. It is very important to address specific drivers of damage to understand why some coral reef patches can maintain the dominance of coral cover while being surrounded by a highly degraded coral ecosystem, although it is known that coral reef degradation and phase shift are multidimensional processes involving multiple states (*McManus & Polsenberg, 2004*; *Bruno et al., 2009*). This apparent patchiness of healthy coral cover throughout the complex, with no evident structure, suggest a loss in the reef complexity with coral reefs tending towards homogenization and becoming a habitat dominated by fleshy macroalgae. The most common species of fleshy macroalgae were *Lobophora* and *Dictyota*, which were also the two species dominating the macroalgae group in *Sánchez et al. (2005)*. Both algae species have shown a negative effect on the fecundity of reef-building corals such as *Orbicella annularis* (*Foster, Box & Mumby, 2008*), which was among the dominant species of hard corals registered by this study and by *Sánchez et al. (2005)* along with the other species of the genus.

In addition, as the initial objective of the scientific expedition was to update the shallow ecosystems maps <30 m we found that areas that were previously reported as *Acropora palmata-Pseudodiploria strigosa* (*Díaz et al., 2000*; *INVEMAR-MINAMBIENTE, 2020*) were replaced by *Pseudodiploria strigosa-Porites astreoides* (*INVEMAR-MINAMBIENTE-DIMAR-CCO, 2021*). In those areas, the contribution of *A. palmata* to the overall coral cover was very low whereas *P. astreoides* was always present and abundant. As shown by our results, *P. astreoides* was one of the most common species of hard corals observed in the stations evaluated, as well as *A. agaricites*, and *S. siderea*. These species are known to cope better with environmental changes (*Estrada-Saldívar et al., 2019*) and stressful agents such as sedimentation (*Cuevas et al., 2009*). For instance, in the Greater Caribbean, there are reports of an increase in the percent cover of *P. astreoides*, usually driven by a cover decline of reef-building species (*Green, Edmunds & Carpenter, 2008*). Therefore, it is important to analyze the community structure composition over time to identify changes related to environmental shifts and detrimental conditions of reefs, and the data provided by us can be replicable to serve as a baseline of information.

Despite not being able to make precise statistical comparisons over time due to the difference in sampling effort with respect to *Sánchez et al. (2005)*, the results obtained through PPI in the four stations suggest relevant changes in the composition of the benthic community. This was validated by the similarity in the percent cover of fleshy macroalgae and hard corals between the two methods employed in the present study (PPI *vs* REA) (Figs. 3B and 3C). Currently, not only fleshy macroalgae are more abundant than hard corals, but differences in cover among both studies are fairly contrasting; for example, we found double macroalgae cover and less than half of the coral cover previously reported (*Sánchez et al., 2005*).

Multiple stressors are radically changing coral reef ecology and functionality, reconfiguring benthic assemblages and populations of various habitat-building coral species. A recent study comparing the nearby atolls of Roncador and Serrana (*Sánchez et al., 2019*) indicated that coral cover demise in Roncador has been replaced with higher

cover of octocorals, with minor change in the abundance of fleshy macroalgae. Nevertheless, their results suggested a generalized deterioration of the reef complex with a possible phase shift to octocorals and algae in Roncador. This is particularly evident within the Caribbean Region in shallow and exposed fore-reef zones, where the diversity and biomass of gorgonian octocorals is higher than scleractinian corals, particularly increasing in the last two decades (*Sánchez, Zea & Díaz, 1998*; *McManus & Polsenberg, 2004*; *Villamizar et al., 2013*; *Ruzicka et al., 2013*). In Quitasueño the coral reef deterioration process looks different and is probably moving faster than in the other two northern atolls of the Seaflower Biosphere Reserve.

## CONCLUSIONS

Overall, our article serves as baseline for current information about isolated reefs in the wider Caribbean. The Rapid Ecological Assessments methodology efficiently allowed us to collect information on over a hundred stations within an extensive reef complex in a short period of time, making this study replicable and comparable for future studies so the reefs can be monitored in time. Coral reefs in Quitasueño seem to be facing a state of fleshy macroalgae dominance, very different from the coral dominance reported 20 years ago. Opportunistic macroalgae such as *Lobophora* and *Dyctiota* are common, and several drivers of coral cover decline such as diseases, aggression, invasion by macroalgae and sponges, and coral tissue predation, among others. We did not see a clear pattern of spatial distribution of stations regarding the percent cover of benthic classes, suggesting that the phase shift is taking place over the entire coral reef complex. In consequence, a significant extension of Quitasueño seems currently just a remnant of the exuberant reefs of yore, which are suffering a gradual and perhaps irreversible decline possibly caused not by a sudden disturbance (*i.e.*, hurricanes) but most likely by a series of chronic stressors. Efforts to understand the reason for the apparent phase shift of Quitasueño should be urgently made, such that coral decline drivers can be mitigated.

## ACKNOWLEDGEMENTS

This expedition was part of an interinstitutional initiative (INVEMAR-HUMBOLT-IDEAM-CIOH-CORALINA). We would like to extend our gratitude to the General Maritime Directorate (DIMAR), the Colombian Ocean Commission (CCO), the Colombian Navy (B/O ARC-Providencia) and CORALINA, for their logistics and support in different aspects of the expedition. We also thank the biologist Irene Arroyave, Juan Carlos Márquez and Juliana Torres for fieldwork support.

### Funding

The Ministry of Environment and Sustainable Development financed this project under the Interadministrative Agreement 628 de 2021. The funders had no role in study design, data collection and analysis, decision to publish, or preparation of the manuscript.

## Grant Disclosures

The following grant information was disclosed by the authors:
The Ministry of Environment and Sustainable Development.
Interadministrative Agreement 628 de 2021.

## Competing Interests

The authors declare that they have no competing interests.

## Author Contributions

- Natalia Rivas conceived and designed the experiments, performed the experiments, analyzed the data, prepared figures and/or tables, authored or reviewed drafts of the article, and approved the final draft.
- Carlos E Gómez conceived and designed the experiments, authored or reviewed drafts of the article, and approved the final draft.
- Santiago Millán conceived and designed the experiments, prepared figures and/or tables, authored or reviewed drafts of the article, and approved the final draft.
- Katherine Mejía-Quintero conceived and designed the experiments, authored or reviewed drafts of the article, and approved the final draft.
- Luis Chasqui conceived and designed the experiments, performed the experiments, authored or reviewed drafts of the article, and approved the final draft.

## Data Availability

    The raw data is available in the Supplemental Files.

## Supplemental Information

Supplemental information for this article can be found online at http://dx.doi.org/10.7717/peerj.15057#supplemental-information.

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
