# Peer review of "Coral reef degradation at an atoll of the Western Colombian Caribbean"

_PeerJ, doi:10.7717/peerj.15057_

## Round 0.1 · original submission · Major Revisions

Dear Authors,

A decision was made on the article submitted to the PEER J Journal, "Coral reef degradation at an atoll of the western Colombian Caribbean". The decision is Major Changes (see attached opinions, consider Reviewers' suggestions).

All modifications and additions are to be completed using tracked changes along with a comments resolution sheet. They are mandatory unless justified by the authors through a message to be evaluated by the editorial team.

In case of disagreement with what was requested by the reviewers, the authors should robustly explain their line of thought and justification for disagreement.

Please feel free to contact me if you have any queries.

Best wishes,
Academic Editor

Reviewer 1 ·

Basic reporting

I would appreciate the authors for evaluating the possible drivers for the extent of degradation of Quitasueño in understanding the process. The manuscript was written according to the journal guidelines. The authors could include more recently published content (the years 2022, 2021, 2020, and 2019) in the introduction and review of the literature section. The authors could directly start explaining the results by avoiding background information in the results and discussion section. The structure of the manuscript falls within the scope of the journal, and I have a few questions and suggestions for improving the need of the manuscript.
I recommend the authors to have a thorough grammar and language check throughout the manuscript as the Quality English score is only 4.6/10 while checked with https://www.aje.com/grammar-check. Hence it’s recommended to reconstruct the English sentences throughout the manuscript. E.g. Line no. 30, spelling mistake in typing “understand”.
The figures were presented well, and I recommend the authors increase the resolution of Figure 2.

Experimental design

The primary research was conducted according to the need and within the scope of the journal. The research questions were explained well, and I feel the experimental protocol could have been performed with more supporting analyses, but the authors have focused only on spatial and temporal scales.
The survey was conducted using REA, and the methodology of this survey was not mentioned in the manuscript or as supporting material.
I could see that the statistical analysis was performed but not evaluated properly. The authors have to look into this carefully. Statistical analysis was not properly designed and executed.
What is about the data reproducibility? The authors have not discussed that.

Validity of the findings

In figure 1, the authors have shown the classification analysis using the clustering approach. The authors need to provide proper statements about this approach. How was this clustering approach performed? What is the outgroup for this study? What is the branch cut-off ratio?
Line: 192-913 The authors claim that “The coral reefs worldwide are in a decline process, which some claim is a phase shift from ecosystems dominated by hard corals to bottoms dominated by fleshy macroalgae”, but the figure does not clearly show this. Proper labelling is essential.
Line 198 -200: Supporting evidence is the highlight in the figure.
How are the distances computed in the clustering analysis? Mention the name of the method.

In figure 2, the authors try to mention the “Spatialized data of the benthic community of Quitasueño”. But the figure was not represented properly. The results are not predictable by observing the figure. Figure resolution must be increased.

Regarding the comparison of statistical methods, the explanation was inadequate. The authors have to explain in detail with appropriate statistical pieces of evidence. Control was not properly defined.

Additional comments

These corrections could be addressed by the authors before publishing in the journal.

Annotated reviews are not available for download in order to protect the identity of reviewers who chose to remain anonymous.

·

Basic reporting

I have carefully read the paper entitled “Coral reef degradation at an atoll of the western Colombian Caribbean”. This manuscript gives an insight into the status of Quitasueño Bank in the Seaflower Biosphere Reserve, Colombia. The manuscript largely aims to describe the status of the Bank through a rapid assessment but also conducted a temporal comparison to assess the ecological changes undergone by reef benthic communities since 2005. The authors suggest that coral cover has declined considerably in the past decades, and fleshy macroalgae cover has increased. Overall, the paper serves as a baseline for current information about isolated reefs in the Caribbean. It gives basic information on the benthic cover and aims to warn of the urgent need for an ecosystem health assessment. I, however, consider the manuscript needs to be substantially improved in the description of the methods and presentation of the results in order to be adequate for publication. Also, the English language should be improved.

Experimental design

1. Survey methods are very superficially described, and therefore it is not possible to assess whether the temporal comparison is valid, or whether the evidence of degradation shown in Figures 4 and 5 is indeed widespread or were sporadic observations during the fieldwork. First, I would suggest expanding the geomorphological description and habitats or zones of the surveyed bank. Some aerial images or reefscape photos of the different areas would help the readers better understand this study's context. Second, please provide more detail about the surveys conducted in 2005, methods, number, and location of the surveyed stations, or at least the depth profile or habitats surveyed by Sánchez et al. (2005). It will be helpful to represent in the map (Figure 2) where the 2005 surveys were conducted. Third, it is essential to demonstrate that the four stations surveyed with PPI are comparable (same stations, depths or habitats) to those conducted in 2005. Please also put the location of these four PPI stations in Figure 2. Four, you also need to treat more carefully the comparison of REA and PPI; this would only be valid when REA and PPI stations were conducted in similar habitat types/depths. For example, if the PPI only included shallow reefs, then data obtained with this method will be only comparable with shallow stations surveyed with the REA.

2. Regarding the methodology is not clear why the authors decided to use two different methods (REA and PPI). Why not only use PPI? Also, if they compare it to another study using the same methodology, why is the sampling effort so different? Do the results of Fig. 3ª and 3b include the same stations? To aid the visual interpretation of the data, I would also suggest having all graphs in figure 3 in the same order and scale.

Validity of the findings

3. As mentioned above, the conclusions from the evidence presented in Figures 4 and 5 must be taken very carefully. These isolated photos without context and a quantitative or semi-quantitative assessment are insufficient to reach any conclusion. Coral overgrown by other organisms or the presence of some predators (e.g. Flamingo Tongue snails) is not indicative of a degradation state. On the contrary, these other organisms are pretty common on Caribbean reefs and are part of the ecological dynamics of these systems. The problem arises when the population of some of these organisms (predators or competitors) drastically increase and threatens large numbers of corals (or other organisms). I, however, did not find any evidence showing this is the case in this manuscript. I would suggest revisiting the data or photos collected in the field and trying to provide more context to these findings—for example, the number of times or stations in which specific “threats” were observed. You could also use the more detailed information obtained in the four PPI stations to provide a quantitative analysis of some of these threats. For example, the proportion of corals affected by a disease or overgrown by another organism.

4. Regarding the report of the presence of the SCTLD, looking at the photos in Fig. 4, it does not seem like SCTLD. The Orbicella annularis in 4a looks more like a white plague, and the Siderastraea looks like a dark spot. A good indicator of SCTLD is that a vast number of colonies of susceptible species, mainly (e.g. Meandrina meandrites, D. cylindrus, P. strigosa, C. natans), have the presence of recent lesions with a high percentage of recent mortality. In figure 5F, there does not seem to be a lot of recently dead colonies, for which I doubt if there is a presence of SCTLD in Quitasueño. I would recommend presenting if they have an approximation of the number of colonies that showed signs of what could be SCTLD. Also, to clarify if Quitasueño has a high abundance of susceptible species to SCTLD? Because this would be a critical alert. I would still recommend using landscape photos as an example where numerous diseased colonies can be seen.

Additional comments

Other comments:

-I recommend improving the introduction since the current structure confuses readers. In lines 54-66, the authors state that Quitasueño is an impacted and threatened reef, but in the next paragraph, authors highlight that it is one of the reefs with high coral cover. I suggest that in lines 48-53, the authors provide a more comprehensive background of Quitasueño reefs, and highlight the main reef-builders, if it was considered a reef in good health before their study. And then describe the studies that address the decline and threats.

-Are the REA and PPI comparable methods? What were the REA photos used for? how many visual estimations were made for each station is not clear—only one or more?

-In the results, the authors say there are no differences between reef zones. Still, it would be interesting to know if, apart from the coral cover, there is a trend between the stations or the groupings from the CLUSTER when assessing the identity of species (coral community composition). That would give important information on the status considering that one of the most drastic changes in coral reefs is defined by the shifts in community composition.

Line 26, the correct name is “Stony Coral Tissue Loss Disease”

Line 35 Change “this” to “the”

Line 38-41. The sentence is difficult to read; I would suggest something like, “Some threats work at a global scale in relation to climate change, such as coral bleaching due to sea surface temperature increases (Brown et al., 2019); others operate at the local or regional level, mainly related to local human activities, such as pollution, terrestrial runoff, and fishing (McLean et al., 2016).

Line 43 Add “also” before reported

Line 84 Change primer to initial

Line 166 Change makeover to shift

Line 103, the Díaz reference is missing.

Line 210-211 Do these stations present any environmental characteristics different from the others that could influence this difference?

The title in Fig 3 b and c should be 2021

·

Basic reporting

The survey report briefly describes the degradation of coral reefs in the Western Colombian Caribbean region. The language of the manuscript can be polished for better readability. The article's introduction briefly describes the deterioration of coral reefs due to various macroalgal species. The survey report also defined the present issues faced by the coral reefs, which inhibit their growth.

The major query is that Coral reefs have a mutualism with macroalgal species, and there is a lack of the mechanism behind the growth of the algae over the corals, which should be elaborated in the introduction.

Experimental design

The survey work meets the scope of the journal. The experimental design concentrates on the spatial and temporal approaches. Various factors are responsible for the deterioration of corals.

What is the concept behind explaining/taking only spatial and temporal approaches to degrade coral reefs?

The Stony Coral Tissue Loss Disease (SCTLD) of coral reefs was already reported in 2014; what is the innovative part of the survey that reveals and adds points to the deterioration of corals?

Validity of the findings

The discussion part needs more elaboration on the spatial and temporal parameters, which were the core part of the degradation of the coral reefs.

The graphical data and the images of the affected coral reefs were clear with supporting results. The conclusion part fulfils the results. The whole survey was explained experimentally with satisfactory figures of good quality with proper labelling.

Additional comments

I recommend minor revisions to this manuscript.

---

## Round 0.2 · accepted · Accept

Congratulations to the authors for improving the article based on the suggested revision!

Reviewer 1 ·

Basic reporting

The authors have revised the paper perfectly.

Experimental design

This section has been elaborated clearly.

Validity of the findings

The results have been validated by the authors in a good way.

Additional comments

The authors have revised the paper well. I can be considered for publication as per the journal's policy.

·

Basic reporting

I have now carefully read the most recent version of paper entitled "Coral reef degradation at an atoll of the western Colombian Caribbean".

I am satisfied with how the authors addressed my previous comments. The methods and results sections substantially improved, and the introduction now gives a more robust background of the study and the locality.

Overall, the manuscript has a better structure and falls within the journal's scope, which is why I now consider it suitable for publication.

I would only encourage the authors to check the references and citations. I believe the citation you provide for Estrada-Saldivar et al. 2019 is incorrect. Not sure if you have a similar issue with other references.

Experimental design

The methods have been expanded and now provide a more detailed description of the field methods and temporal comparison.

Validity of the findings

Results es are now presented clearly, and the analyses are sound.

·

Basic reporting

The authors have revised the paper well.

Experimental design

This section is improved now.

Validity of the findings

The authors have explained the findings perfectly.

Additional comments

The paper can be considered for publication now.